# OpenReview forum: "Jailbreaking Vision-Language Models Through the Visual Modality"
_ICML.cc/2026/Conference — ICML 2026 regular_

### Official Review · Reviewer_6oEd · 2026-03-09

**Soundness:** 3
**Presentation:** 3
**Significance:** 3
**Originality:** 3
**Overall Recommendation:** 4
**Confidence:** 4

**Summary:**

The paper argues that the visual modality is an effective jailbreak channel for frontier vision-language models (VLMs). It proposes four attack strategies—visual cipher, visual object replacement, visual text replacement, and visual analogy riddle—and evaluates them on five production VLMs.   The authors use prompt neutralization to test whether harmful intent can be reconstructed from images.   Evaluation uses a three-LLM judge panel with a 0–3 rubric and defines attach success as Best-of-5 yielding a 3 (“Compliance”).

**Compliance With Llm Reviewing Policy:**

Affirmed.

**Final Justification:**

My main concerns were about evaluation rigor, especially the use of Best-of-5 as the primary metric, lack of statistical reporting, absence of human grounding for the judge rubric, and limited failure-mode analysis. The rebuttal addressed these points well: the added K=1/K=5 analysis clarifies the inflation issue, the planned statistical reporting would strengthen the quantitative claims, and the rubric-distribution breakdown gives a much clearer picture of mechanism-specific failures. Overall, the rebuttal reinforces my original assessment at Weak Accept.

**Key Questions For Authors:**

Please clarify the weakness I pointed out in the limitations section

**Limitations:**

No.

The paper notes some limitations (judge ambiguity, dependence on T2I quality, dual-use concerns), but it should additionally (i) explicitly discuss how Best-of-5 inflates success relative to single-shot use, (ii) include uncertainty/statistical reporting for key comparisons, (iii) add a small human validation subset for the judge rubric, and (iv) provide qualitative failure-mode analysis to clarify mechanisms and mitigation implications.

**Strengths And Weaknesses:**

## **Strengths**

1. **Important framing**: The paper highlights “vision as an attack surface,” a practically relevant safety concern for deployed VLMs.
2. **Clear taxonomy of visual jailbreak mechanisms**: The four attacks cover distinct obfuscation/indirection mechanisms, making the contribution easy to interpret.
3. **Prompt neutralization is a useful diagnostic tool**: Slot-based neutralization is a reasonable attempt to isolate meaning reconstruction from direct harmful prompting.
4. **Mitigation direction is included**: The output-side guardrail experiment improves practical relevance.

---

## **Weaknesses**

1. **No statistical testing / uncertainty reporting**: The paper reports success rates across models/attacks but does not provide confidence intervals, significance tests, or variance estimates. Given the multi-model comparisons and the Best-of-5 evaluation design, uncertainty reporting is important for interpreting differences robustly.
2. **No human validation of the automated evaluation**: Evaluation relies on agreement among multiple LLM judges. High inter-LLM agreement only indicates consistency with each other; it does not guarantee correctness, and the judges may share similar biases or safety standards. Therefore, a human-labeled grounding subset (and judge–human agreement) is needed to validate the rubric decisions, especially around borderline cases.
3. **Primary success criterion can inflate effectiveness (analogous to permissive ASR)**: Success is defined as Best-of-5 producing a level-3 (“Compliance”) output at least once. This can materially inflate attack success relative to single-shot deployment and can overemphasize occasional failures of guardrails. The paper would be stronger if it foregrounded single-shot results and reported distributions over rubric levels (0–3), not only a Best-of-5 success indicator.
4. **Limited qualitative analysis / failure-mode understanding**: The paper describes the attacks, but provides limited qualitative insight into *why* attacks succeed or fail across models (e.g., what visual cues or model behaviors drive compliance), and lacks a systematic failure taxonomy. Without qualitative analysis, it is difficult to validate the proposed mechanisms (e.g., “meaning reconstruction from vision”) and to derive actionable guidance for defenses beyond the tested guardrail.

---

> ### Author Rebuttal · Authors · 2026-03-30
>
> > **W1.** No statistical testing / uncertainty reporting.
>
> Our reporting follows the convention of established benchmarks [1, 2, 3], which report ASR without confidence intervals or significance tests. We computed full statistics and will add them in the camera-ready: (a) K=1 single-shot ASR alongside K=5 (W3), (b) 95% Wilson confidence intervals, and (c) Fisher's exact tests with Benjamini-Hochberg correction for key visual-vs-textual comparisons.
>
> > **W2.** No human validation.
>
> We agree that inter-LLM agreement is not a substitute for human grounding. While working on our methods, we manually evaluated hundreds of examples, and we presented qualitative examples in the paper. We will annotate a 100-response stratified subset across attack types, rubric levels, and model families, reporting judge-human agreement in camera-ready.
>
> > **W3.** Best-of-5 inflates effectiveness.
>
> Best-of-K is standard in jailbreak research [1, 3] and reflects realistic attacker practice. HarmBench [1] uses single-attempt greedy decoding; [3] reports best-of-N as primary metric. K=1 / K=5 ASR (%) from existing outputs:
>
> |Attack|Haiku 4.5|Gemini Flash 3|GPT 5.2|Qwen 235B|Qwen 32B|Gemini 3.1 Pro|
> |-|-|-|-|-|-|-|
> |V. Cipher|15.1/40.9|90.6/97.5|3.8/8.2|62.9/86.2|62.3/87.4|8.2/14.5|
> |V. Obj. Repl.|0.7/4.1|34.5/52.0|6.8/11.5|30.1/35.6|28.1/41.1|31.0/45.6|
> |V. Text Repl.|3.6/12.9|14.9/32.8|9.4/14.4|30.6/51.5|36.8/58.1|32.6/48.6|
> |V. Riddle|13.8/13.8|52.2/52.2|13.2/13.2|29.6/29.6|38.4/38.4|6.3/6.3|
> |T. Cipher|4.4/10.7|74.8/89.3|1.9/5.7|70.4/86.8|58.5/84.9|7.5/15.1|
> |T. Repl.|6.1/8.1|55.4/58.8|12.8/16.9|25.3/29.5|35.6/39.0|18.4/19.0|
> |T. Riddle|39.6/39.6|67.9/67.9|24.5/24.5|51.6/51.6|62.3/62.3|17.0/17.0|
>
> Inflation varies by mechanism: Visual Cipher up to 2.7x (each seed produces different glyph assignments); Visual Object Replacement - most models 1.2–1.7x, Haiku is an outlier at 5.9x due to very low K=1 (same images, repeated queries); Visual Text Replacement up to 3.6x (same images, repeated queries); Visual Analogy Riddle 1.0x (all riddle combinations are ranked beforehand, so the strongest is always tried first). The camera-ready will include K=1 & K=5 with full rubric distributions (levels 0–3) for every attack/model pair.
>
> > **W4.** Limited failure-mode understanding.
>
> Sec. 5.2 and the Apdx discuss where and why attacks fail. Below we show full rubric distributions across all K=5 attempts (Refusal/Misunderstanding/Partial/Compliance %):
>
> **Cipher attacks:**
>
> |Attack|Haiku 4.5|Flash 3|GPT 5.2|Qwen 235B|Qwen 32B|Gemini 3.1 Pro|
> |-|-|-|-|-|-|-|
> |V. Cipher|45.8/37.2/3.5/13.5|5.9/2.5/0.6/90.9|93.1/0.4/1.1/5.4|19.7/16.7/2.6/60.9|20.3/12.8/5.3/61.6|76.0/2.1/14.5/7.4|
> |T. Cipher|95.2/0.6/0.3/3.9|18.9/2.8/0.9/77.5|97.2/0.1/0.5/2.1|18.2/5.0/3.0/73.7|33.0/2.0/5.5/59.5|77.0/1.5/11.4/10.1|
>
> **Object/Text Repl. attacks:**
>
> |Attack|Haiku 4.5|Flash 3|GPT 5.2|Qwen 235B|Qwen 32B|G3.1 Pro|
> |-|-|-|-|-|-|-|
> |V. Obj. Repl.|83.0/13.2/0.5/3.2|17.7/39.5/4.2/38.6|58.4/29.6/1.6/10.4|1.6/64.2/4.9/29.2|1.9/62.6/4.5/31.0|29.0/42.9/1.5/26.6|
> |V. Text Repl.|72.3/19.7/2.0/6.0|68.1/10.9/2.1/18.9|78.7/9.4/1.8/10.1|16.1/49.8/7.9/26.2|12.9/45.5/8.6/33.0|56.4/9.9/3.2/30.5|
> |T. Repl.|80.9/11.6/1.6/5.8|21.4/21.5/1.8/55.4|79.7/10.1/1.4/8.8|12.8/59.9/1.2/26.1|12.4/48.2/2.4/36.9|50.8/5.7/33.8/9.7|
>
> **Riddle attacks**
>
> |Attack|Haiku 4.5|Flash 3|GPT 5.2|Qwen 235B|Qwen 32B|G3.1 Pro|
> |-|-|-|-|-|-|-|
> |V. Riddle| 25.8/67.8/1.0/5.4 | 16.0/47.5/0.9/35.6 | 60.3/33.3/1.3/5.2 | 12.4/70.4/0.5/16.7 | 7.7/75.1/0.1/17.0 | 82.8/15.8/0.1/1.3 |
> |T. Riddle| 37.9/37.1/2.8/22.2 | 12.3/33.0/2.1/52.6 | 58.0/26.7/1.8/13.4 | 27.1/38.1/1.0/33.8 | 11.2/44.0/0.4/44.4 | 87.9/8.3/0.4/3.3 |
>
> The distributions show mechanism-specific failure patterns:
>
> - **Cipher**: GPT-5.2 refuses 93–97% (detects the cipher); Flash complies 78–91% (weak cipher resistance). Haiku: 45.8% refusal + 37.2% misunderstanding for visual but 95.2% refusal for textual - the visual modality specifically weakens defenses.
> - **Object Replacement**: Qwen shows 63–64% misunderstanding (fails to recover the original referent from the substituted object); Gemini 3.1 Pro shows 42.9% misunderstanding yet 26.6% compliance.
> - **Visual Text Replacement**: Qwen exhibits 45-49% misunderstanding + 7–8% partial compliance - a gray zone of partial intent decoding.
> - **Riddle**: For most models, dominated by misunderstanding (up to 75% visual), not refusal; GPT-5.2 and Gemini 3.1 Pro are the exceptions where refusal is the primary failure mode.
>
> The camera-ready will add representative examples per rubric level and further defense-design observations.
>
> > **Limitations.**
>
> We will address all four points in the camera-ready: K=1 vs K=5 reporting with inflation analysis, Wilson CIs and Fisher's exact tests, 100-response human annotation, and expanded failure-mode analysis.
>
> **Ref**
>
>
> [1] Mazeika et al., "HarmBench: [...]", ICML 2024.
>
> [2] Chao et al., "JailbreakBench: [...] ", NeurIPS 2024.
>
> [3] Hughes et al., "Best-of-N Jailbreaking", NeurIPS 2025.

---

> > ### Author Rebuttal · Reviewer_6oEd · 2026-04-02
> >
> > The response addresses most of my concerns, especially through the added K=1/K=5 analysis, the planned statistical reporting, and the rubric-distribution/failure-mode breakdowns. My only remaining minor concern is that the proposed 100-response human validation is still relatively small, so I would encourage the authors to report the sampling protocol clearly and include judge-human agreement in the final version. That said, the rebuttal largely resolves the issues I had raised contingent on this proposed human validation being conducted and the other additions to the paper being made.

---

### Official Review · Reviewer_QMEj · 2026-03-12

**Soundness:** 4
**Presentation:** 4
**Significance:** 2
**Originality:** 3
**Overall Recommendation:** 5
**Confidence:** 4

**Summary:**

The paper introduces four new jailbreaking attacks for VLMs.
They are (1) a visual cipher that encodes
harmful instructions as sequences of symbols that the
model must decode using a legend, (2) visual object
replacement, where a harmful object in an image is replaced
with a benign one, (3) visual text replacement, where
harmful words embedded in images are replaced with benign
placeholders, and (4) visual analogy
riddles, which encode harmful concepts implicitly through
visual analogy puzzles that the model solves.

The empirical analysis is thorough and careful. They demonstrate that their attacks are effective against a range of frontier LLMs (albeit with different performance between the methods and models).

**Compliance With Llm Reviewing Policy:**

Affirmed.

**Final Justification:**

I maintain my current score of accept. Great work!

**Key Questions For Authors:**

# Questions

Q1: Can you expand on the performance of the textual versions of the attacks? Does this warrant a slight reframing of the paper?

Q2: Would it be possible to test the attack methods against larger frontier models, e.g. gemini 3, gpt 5, Opus 4.6?

**Limitations:**

Yes, limitations are discussed in detail!

**Strengths And Weaknesses:**

## Strengths

* Redteaming VLMs is an important area of work, making the paper significant.
* The specifics of the attacks used, are to my knowledge, novel. While prior papers have used the visual inputs of VLMs to jailbreak models, I have not seen the specific four attacks presented in this work.
* The presentation is very good. The paper is easy to follow, and figures explain the different attack types well.
* Perhaps the largest strength of the work is the rigor of the empirical analysis. The authors are careful to analyze the performance of their LLM judge. They use a pessimistic ensemble that makes me trust the empirical findings more.


## Weaknesses

* Although the paper is framed around vision based attack, in Table 2, the textual versions seem to be very effective also, and in fact outperform the visual version in many areas. These attacks are still novel, however it draws into question the visual framing of the paper. You have some discussion on line 267 for visual riddle, but I think more discussion would be valuable.
* The attacks in the paper could be seen as incremental changes from the wealth of other vision language attack papers. With that being said, the black-box attack vector and attack success rates are compelling.

---

> ### Author Rebuttal · Authors · 2026-03-30
>
> We thank the reviewer for the positive assessment, particularly noting the "excellent soundness," "very good presentation," and the "rigor of the empirical analysis" including the pessimistic judge ensemble.
>
> > **W1.** Although the paper is framed around vision-based attack, in Table 2, the textual versions seem to be very effective also, and in fact outperform the visual version in many areas. These attacks are still novel, however it draws into question the visual framing of the paper.
>
> The effectiveness of textual counterparts is not evidence against the paper; it is part of its message. The same hidden-intent mechanisms work across modalities, and current safety tuning does not focus on the visual modality to the best of our knowledge. The paper's thesis is that the visual modality is a neglected attack surface, not that it is universally stronger. Some mechanisms benefit more from visual indirection (Visual Cipher: 40.9% vs 10.7% on Claude Haiku), while others currently benefit from the greater clarity of text (Analogy Riddle). We will reframe the camera-ready around the cross-modal alignment gap rather than visual superiority.
>
> > **W2.** The attacks in the paper could be seen as incremental changes from the wealth of other vision language attack papers. With that being said, the black-box attack vector and attack success rates are compelling.
>
> Our attacks differ from standard typographic or direct-image attacks: the model must reconstruct hidden harmful intent from visual indirection, substitution, or analogy, rather than simply read explicit harmful content from an image. To contextualize, we ran prior VLM jailbreak baselines (TYPO, SD, SD+TYPO from MM-SafetyBench [4], HADES [5], FigStep [6], and PAIR [7]) under our HarmBench evaluation with the same judging mechanism:
>
> | Attack | Haiku4.5 | GeminiFlash 3 | GPT5.2 | Qwen235B | Qwen32B | Gemini 3.1 Pro |
> |---|---:|---:|---:|---:|---:|---:|
> | **Textual Cipher** | 10.7 | 89.3 | 5.7 | 86.8 | 84.9 | 15.1 |
> | **Visual Cipher** | 40.9 | 97.5 | 8.2 | 86.2 | 87.4 | 14.5 |
> | **Textual Replacement** | 8.1 | 58.8 | 16.9 | 29.5 | 39.0 | 19.0 |
> | **Visual Obj. Repl.** | 4.1 | 52.0 | 11.5 | 35.6 | 41.1 | 45.6 |
> | **Visual Text Repl.** | 12.9 | 32.8 | 14.4 | 51.5 | 58.1 | 48.6 |
> | **Textual Riddle** | 39.6 | 67.9 | 24.5 | 51.6 | 62.3 | 17.0 |
> | **Visual Riddle** | 13.8 | 52.2 | 13.2 | 29.6 | 38.4 | 6.3 |
> | TYPO [4] | 5.0 | 11.9 | 5.7 | 33.3 | 37.7 | 5.0 |
> | SD [4] | 6.3 | 22.2 | 10.8 | 48.7 | 56.3 | 7.6 |
> | SD+TYPO [4] | 11.5 | 20.3 | 6.1 | 44.6 | 60.8 | 6.8 |
> | HADES [5] | 9.0 | 12.0 | 2.0 | 11.0 | 32.0 | 13.1 |
> | FigStep [6]| 45.9 | 10.1 | 3.8 | 49.1 | 11.3 | 10.1 |
> | PAIR [7] | 3.8 | 16.4 | 4.4 | 18.9 | 19.5 | 15.7 |
> | PAIR [7] (full, N=30) | 49.1 | 86.8 | 40.9 | 91.8 | 88.7 | 76.7 |
>
> PAIR has two variants: a **budget-matched** version (5 target queries, no attacker LLM) and a **full** version (N=30, \~120 target queries + attacker LLM). Prior baselines are weak or model-selective: HADES averages 13.2%; FigStep works mainly on Claude Haiku and Qwen-235B. At equal query budget, Visual Cipher outperforms PAIR on 5 of 6 models. With full budget (~24x more queries plus a separate LLM that iteratively refines attack prompts), PAIR surpasses Visual Cipher on 5 of 6 models, but our attacks require no optimization loop or auxiliary model.
>
>
> > **Q1.** Can you expand on the performance of the textual versions of the attacks? Does this warrant a slight reframing of the paper?
>
> We appreciate this observation and agree that the framing deserves tightening. Our abstract states that visual attacks "achieve comparable and sometimes superior success rates," and Sec 5.2 is explicit about where text outperforms visual.
> The core thesis is not "visual > text" but that vision is an independently exploitable attack surface:
>   - Visual Cipher achieves 40.9% on Claude-Haiku vs. 10.7% textual - a nearly 4x gap on a safety-hardened model.
>   - Visual Text Replacement outperforms text on 3 of 5 models (e.g., Qwen-235B: 51.5% vs. 29.5%).
>   - Where visual underperforms (Analogy Riddle), the cause is perceptual ambiguity, not stronger safety enforcement: misunderstanding rates are substantially higher for visual riddles (Appendix). This might change with upcoming models with stronger visual reasoning capabilities.
>
> The revised framing will state that: (1) visual attacks are not uniformly stronger, (2) the gap is mechanism- and model-dependent, and (3) the key finding is the existence of a distinct visual route for safety bypass, not a universal visual advantage.
>
>
> > **Q2.** Would it be possible to test the attack methods against larger frontier models, e.g. Gemini 3, GPT 5, Opus 4.6?
>
> We added **Gemini 3.1 Pro** to our evaluation and compared the effectiveness of our attack to other baseline attacks. Please see the answer for Weakness 2.
>
>
> **References**
>
> Due to space constraints, we refer to the response to y8TN for actual references.
>
> ---

---

> > ### Author Rebuttal · Reviewer_QMEj · 2026-04-03
> >
> > Thank you for taking the time to address my concerns! I agree with the new changes to the abstract. Overall I will keep my current recommendation of accept, great work!

---

### Official Review · Reviewer_Qwye · 2026-03-13

**Soundness:** 2
**Presentation:** 2
**Significance:** 2
**Originality:** 2
**Overall Recommendation:** 3
**Confidence:** 4

**Summary:**

This paper systematically investigates a critical security vulnerability in Vision-Language Models (VLMs): the visual modality can be independently exploited as a carrier for jailbreak attacks. To demonstrate this, the authors propose four novel visual jailbreak methods. The common core of these methods is to encode or imply malicious intent through visual structure, context, or reasoning processes, while keeping the surface-level text and image content "benign."
The authors conduct a comprehensive evaluation on five popular VLMs. The results show that the success rates of visual attacks are generally comparable to, and in some models even higher than, those of text-only attacks. This reveals a clear cross-modal alignment gap: safety training based on text fails to transfer effectively to visual representations. The paper also provides preliminary interpretability analysis, suggesting that attacks may exploit a temporal mismatch between safety mechanisms and semantic understanding. Finally, the authors note that lightweight output-side classifiers can effectively defend against such attacks, offering a feasible mitigation strategy for practical deployment.

**Compliance With Llm Reviewing Policy:**

Affirmed.

**Key Questions For Authors:**

See weaknesses

**Limitations:**

Yes

**Strengths And Weaknesses:**

## Strengths
- Creative and Diverse Attack Methods:​ The four proposed attack methods cover different visual encoding and hinting strategies, ranging from direct symbol decoding to object replacement and analogy puzzles that rely more on contextual reasoning, demonstrating the breadth of the visual attack surface. The Visual Analogy Riddle, in particular, which exploits the model's abstract reasoning capability, is a novel form of attack.
- Benchmarking and Evaluation:​ The evaluation is based on HarmBench and employs a "prompt neutralization" strategy, effectively isolating the contribution of the visual modality for a fairer comparison.
Broad Model Coverage:​ The evaluation includes five state-of-the-art models, both closed-source and open-source, making the conclusions representative.
## Weaknesses
- Imbalance between breadth and depth of discussion:​ As noted by the reviewers, the paper expands in multiple directions, but the exploration in each direction feels somewhat superficial. The overall writing gives an impression of being cursory, lacking in-depth analysis.
Lack of a unified classification framework and motivation for the attack methods:​ The paper presents four attack methods in parallel but does not explain how these methods were derived or summarized, or whether there are more fundamental connections or dimensions between them. This creates an impression of being somewhat ad-hoc, with a slightly loose structure. The reviewer's question about "whether other methods exist" is valid. The paper fails to position its work within a higher-level attack classification framework, thereby limiting the systematic nature and theoretical depth of the research.
- Questionable definition of "harmfulness" and appropriateness of examples:​ The reviewer astutely pointed out the issue with the "bomb → banana" example. This critique touches on a fundamental difficulty in VLM safety evaluation: there is an essential difference in risk level between "describing a harmful object" and "instructing a harmful action." An image of a bomb itself may exist in legitimate film, news, or educational materials. The paper's classification of a model recognizing such an image and associating it with the word "bomb" as a "successful jailbreak" may overestimate its actual harm and blur the boundary of a "safety violation." This weakens the persuasiveness of some attack examples, making them seem more like tests of the model's visual recognition and contextual association capabilities rather than inductions to produce practical malicious behavior.

---

> ### Author Rebuttal · Authors · 2026-03-30
>
> We thank the reviewer for the valuable feedback and for recognizing the  *creative and diverse attack methods*  covering different visual encoding strategies, the HarmBench-based benchmarking with prompt neutralization, and the *broad model coverage*.
>
> > **W1.** Imbalance between breadth and depth of discussion. The paper presents four attack methods in parallel but does not explain how these methods were derived or summarized, or whether there are more fundamental connections or dimensions between them. This creates an impression of being somewhat ad-hoc. The reviewer's question about "whether other methods exist" is valid. The paper fails to position its work within a higher-level attack classification framework.
>
> We appreciate this point and agree the unifying structure should be made clearer and the exposition deeper. The attacks share a common principle: **all four preserve benign surface appearance while shifting the burden of harmful-intent recovery onto visual reasoning**. In the revision, we will introduce an organizing framework based on the question: *How can prohibited semantics be recovered from visually benign surface content?*
>
> Under this view, the attacks fall into three mechanism-level categories:
> - **Symbolic decoding from visual structure**: Visual Cipher
> - **Contextual referent recovery under substitution**: Visual Object Replacement and Visual Text Replacement
> - **Compositional inference from benign components**: Visual Analogy Riddle
>
> This is a partial taxonomy, not a closed list. Other visual encoding mechanisms likely exist (e.g., steganographic embedding, spatial arrangement cues). We see our contribution as the first systematic evidence that multiple distinct visual indirection mechanisms can bypass safety training, and we will frame it accordingly.
>
> > **W2.** Questionable definition of "harmfulness" and appropriateness of examples. The "bomb to banana" example: there is an essential difference between "describing a harmful object" and "instructing a harmful action." The paper's classification of a model recognizing such an image and associating it with the word "bomb" as a "successful jailbreak" may overestimate its actual harm.
>
> We believe that there is some misunderstanding from the reviewer side and want to clarify that this is not how our evaluation works. **Our evaluation does not count object recognition as a successful jailbreak.** A response is scored as level 3 (Compliance) only if the model produces materially useful harmful assistance for the full HarmBench behavior. Recognizing that a banana substitutes for "bomb" is not counted as success. The model must go further and produce actionable harmful guidance for the downstream task (e.g., detailed operational steps for smuggling explosives). This is enforced by both the HarmBench rubric and our conservative 3-judge ensemble, which requires majority agreement at the highest severity level.
>
> We agree this distinction should be made more explicit in the paper and will add annotated examples in the revision separating: recognition only (not a jailbreak), partial harmful assistance, and full compliance (the only case counted as success).
>
> ---

---

> > ### Author Rebuttal · Reviewer_Qwye · 2026-04-03
> >
> > Thank you for the response. Regarding the first point (W1), I appreciate the effort to provide a unifying framework. However, I feel there may be a slight misunderstanding of my original concern.
> >
> > My point was about the derivation process behind the selection of the four specific methods. The principle of “shifting harmful intent to the visual modality” is indeed the paper’s premise. My question is: how did the authors systematically arrive at these four attack types, rather than other plausible ones?
> >
> > Without a clear explanation of the methodological reasoning—whether based on a taxonomy of visual perturbations, a search over a design space, or some other generative process—the choice of methods can appear somewhat arbitrary. It leaves open the question of whether other potentially effective visual attacks remain unexplored, and why these four in particular were selected for in-depth study.
> >
> > To strengthen the methodological grounding, it would be helpful if the authors could briefly clarify the process behind the generation and selection of these attacks. This would address concerns about the completeness and systematicity of the exploration, and help readers better understand the scope and contribution of the work.

---

> > > ### Author Response · Authors · 2026-04-04
> > >
> > > We thank the Reviewer for the follow-up response. We would like to note that our rebuttal addressed both Weaknesses 1 and 2, yet the Reviewer’s acknowledgment only engages with Weakness 1. We would appreciate the Reviewer’s response to our clarification on Weakness 2 as well.
> > >
> > > On the derivation process: there was no formal taxonomy or design space enumeration. We came up with the proposed attacks through good ol’ human creativity. We 1) initially noticed that VLM safety training relatively neglects vision; 2) asked ourselves "what visual structures could encode harmful intent while looking benign?”; and 3) iteratively invented and refined attacks that exploit qualitatively different visual reasoning capabilities (symbol decoding, object substitution, text context, analogical inference). We note that this is how most novel attack vectors in security research are discovered. We believe that creative exploration precedes systematization.
> > >
> > > We never claimed that our four proposed attacks exhaust this space. Our contribution is the first *evidence* that multiple distinct visual indirection mechanisms bypass safety training. Requiring a formal generative methodology for discovering novel attacks in security research would be a requirement that few empirical security papers fulfill.

---

### Official Review · Reviewer_y8TN · 2026-03-15

**Soundness:** 2
**Presentation:** 3
**Significance:** 3
**Originality:** 3
**Overall Recommendation:** 4
**Confidence:** 3

**Summary:**

This paper studies jailbreak attacks on VLMs through the visual modality. The main idea is that harmful intent can be encoded through visual tokens while keeping the text modality relatively benign. The paper proposes four attack types: 1) Visual Cipher, which encodes harmful instructions as visual glyphs with a decoding legend, 2) Visual Object Replacement, which replaces harmful objects with benign substitutes while preserving scene context, 3) Visual Text Replacement, which replaces harmful text inside images with benign placeholders while keeping visual context, and 4) Visual Analogy Riddle, which decomposes harmful intent into benign visual riddles whose combined solution recovers the unsafe concept. The paper evaluates these attacks on five VLMs, using HarmBench behaviors, a three-model judge ensemble, and Best-of-5 evaluation.

**Compliance With Llm Reviewing Policy:**

Affirmed.

**Final Justification:**

Adjusting score due to rebuttal.

**Key Questions For Authors:**

1. The paper claims that visual attacks “match or exceed” text-only baselines, but Table 2 shows a more mixed picture, especially for visual analogy and some object-replacement settings, where the text baselines are better than visual attacks. Can the authors clarify the exact scope of this claim?


2.Questions from weaknesses sections.

**Limitations:**

yes

**Strengths And Weaknesses:**

Strengths:


1. The empirical study is fairly extensive: five VLMs, with proposed four types of visual modality attack, and a conservative judge ensemble.


2. The attack design is reasonably broad. Instead of only one attack type, the paper covers four different mechanisms.




Weaknesses:



1.The high-level claims feels somehow not strong. For example, in Table2, visual analogy is consistently weaker than textual analogy across all models, and visual object replacement is not uniformly stronger than the text baseline. Does it mean the text modality modification is strong enough, or there are some condition patterns that the attack from text modality is better/easier than visual modality.

2.What about the other attack baselines? The paper only presents the different proposed attack scenarios, but did not compare with SOTA attack baselines.

---

> ### Author Rebuttal · Authors · 2026-03-30
>
> We thank the reviewer for the careful reading and for noting the strengths of our work, including the "extensive empirical study" covering five VLMs and four attack types, and the " broad" attack design.
>
> > **W1.** The high-level claims feels somehow not strong. For example, in Table 2, visual analogy is consistently weaker than textual analogy across all models, and visual object replacement is not uniformly stronger than the text baseline. Does it mean the text modality modification is strong enough, or there are some condition patterns that the attack from text modality is better/easier than visual modality.
>
> We appreciate this observation and agree that the framing deserves tightening. Our abstract states that visual attacks "achieve comparable and sometimes superior success rates," and Sec 5.2 is explicit about where text outperforms visual.
> The core thesis is not "visual > text" but that vision is an independently exploitable attack surface:
>   - Visual Cipher achieves 40.9% on Claude-Haiku vs. 10.7% textual -- a nearly 4x gap on a safety-hardened model.
>   - Visual Text Replacement outperforms text on 3 of 5 models (e.g., Qwen-235B: 51.5% vs. 29.5%).
>   - Where visual underperforms (Analogy Riddle), the cause is perceptual ambiguity, not stronger safety enforcement: misunderstanding rates are substantially higher for visual riddles (Appendix). This might change with upcoming models with stronger visual reasoning capabilities.
> Per R-QMEj's suggestion, the camera-ready will reframe around the cross-modality alignment gap rather than visual superiority.
>
>
> > **W2.** What about the other attack baselines? The paper only presents the different proposed attack scenarios, but did not compare with SOTA attack baselines.
>
> The paper's primary goal is not to propose the strongest possible attack, but to put a spotlight on the vision modality as a neglected attack surface. Our baselines are deliberately matched text-only counterparts that isolate the modality contribution. Comparing against methods with fundamentally different threat models (e.g., iterative prompt optimization with an attacker LLM) conflates attack strength with modality contribution.
>
> However, we agree that contextualizing against prior VLM attacks adds value. To address this concern, we ran additional SOTA baselines under a matched HarmBench evaluation (best-of-5).
>
> | Attack | Claude Haiku 4.5 | Gemini Flash 3 | GPT 5.2 | Qwen 235B | Qwen 32B | Gemini 3.1 Pro |
> |---|---:|---:|---:|---:|---:|---:|
> | **Ours: Textual Cipher** | 10.7 | 89.3 | 5.7 | 86.8 | 84.9 | 15.1 |
> | **Ours: Visual Cipher** | 40.9 | 97.5 | 8.2 | 86.2 | 87.4 | 14.5 |
> | **Ours: Textual Replacement** | 8.1 | 58.8 | 16.9 | 29.5 | 39.0 | 19.0 |
> | **Ours: Visual Obj. Repl.** | 4.1 | 52.0 | 11.5 | 35.6 | 41.1 | 45.6 |
> | **Ours: Visual Text Repl.** | 12.9 | 32.8 | 14.4 | 51.5 | 58.1 | 48.6 |
> | **Ours: Textual Riddle** | 39.6 | 67.9 | 24.5 | 51.6 | 62.3 | 17.0 |
> | **Ours: Visual Riddle** | 13.8 | 52.2 | 13.2 | 29.6 | 38.4 | 6.3 |
> | TYPO [4] | 5.0 | 11.9 | 5.7 | 33.3 | 37.7 | 5.0 |
> | SD [4] | 6.3 | 22.2 | 10.8 | 48.7 | 56.3 | 7.6 |
> | SD+TYPO [4] | 11.5 | 20.3 | 6.1 | 44.6 | 60.8 | 6.8 |
> | HADES [5] | 9.0 | 12.0 | 2.0 | 11.0 | 32.0 | 13.1 |
> | FigStep [6]| 45.9 | 10.1 | 3.8 | 49.1 | 11.3 | 10.1 |
> | PAIR [7] | 3.8 | 16.4 | 4.4 | 18.9 | 19.5 | 15.7 |
> | PAIR [7] (full, N=30) | 49.1 | 86.8 | 40.9 | 91.8 | 88.7 | 76.7 |
>
> PAIR has two variants: a **budget-matched** version (5 target queries, no attacker LLM) and a **full** version (N=30, \~120 target queries + attacker LLM). Prior baselines are weak or model-selective: HADES averages 13.2%; FigStep works mainly on Claude Haiku and Qwen-235B.  At equal query budget, Visual Cipher outperforms PAIR on 5 of 6 models. With full budget (~24x more queries plus a separate LLM that iteratively refines attack prompts), PAIR surpasses Visual Cipher on 5 of 6 models, but our attacks require no optimization loop or auxiliary model.
>
> > **Q1.** The paper claims that visual attacks "match or exceed" text-only baselines, but Table 2 shows a more mixed picture. Can the authors clarify the exact scope of this claim?
>
> Our main contribution is to highlight that the visual channel is a distinct, under-protected route for hidden-intent reconstruction. We will tighten the language in the camera-ready.
>
> **References**
>
> [4] Liu et al., "MM-SafetyBench: A Benchmark for Safety Evaluation of Multimodal Large Language Models", ECCV 2024.
>
> [5] Li et al., "Images are Achilles' Heel of Alignment: Exploiting Visual Vulnerabilities for Jailbreaking Multimodal Large Language Models", ECCV 2024.
>
> [6] Gong et al., "FigStep: Jailbreaking Large Vision-Language Models via Typographic Visual Prompts", AAAI 2025.
>
> [7] Chao et al., "Jailbreaking Black Box Large Language Models in Twenty Queries", IEEE SaTML 2025.
>
> ---

---

> > ### Author Rebuttal · Reviewer_y8TN · 2026-04-04
> >
> > Thanks for the rebuttal. I have increased my score accordingly.

---

### Decision · Program_Chairs · 2026-04-30

**Decision:**

Accept (regular)

**Comment:**

This paper studies visual-modality jailbreak attacks for VLMs and proposes four diverse attack mechanisms. The empirical study is strong with broad attack coverage and careful evaluation.  Main concerns were mixed evidence for visual over text attacks, limited stronger baselines, limited unifying structure for the four attacks, and evaluation issues such as Best-of-5, missing statistical reporting, and limited human validation. The rebuttal addresses most of these concerns effectively, and presents a clear, empirically strong contribution to VLM safety.